# Depressive symptoms and associated factors among persons with physical disabilities in disability care homes of Kathmandu district, Nepal: A mixed method study

Prabin Karki[1]*, Prasant Vikram Shahi[2], Krishna Prasad Sapkota[3], Rabindra Bhandari[1,4], Nabin Adhikari[1,5], Binjwala Shrestha[1]

1 Institute of Medicine, Tribhuvan University, Kathmandu, Nepal, 2 Community and Global Health, Graduate School of Medicine, The University of Tokyo, Tokyo, Japan, 3 Karuna Foundation Nepal, Kathmandu, Nepal, 4 Nepal Health Research Council, Ramshahpath, Kathmandu, Nepal, 5 Department of Research and Development Division, Dhulikhel Hospital, Dhulikhel, Kavrepalanchowk, Nepal

* prabinkarki8247@gmail.com

**Data Availability Statement:** All data are in the manuscript and/or supporting information files.

## Abstract

Depression is one of the most common mental disorders, affecting 300 million people worldwide and 75% of these occur in low- and middle-income countries. Persons with physical disabilities are vulnerable groups and are more prone to experience depressive symptoms than the general population. This study investigated the prevalence of depressive symptoms and the associated factors among persons with a physical disability. We conducted a concurrent triangulation mixed methods design using Beck's Depression Inventory scale among 162 persons with physical disabilities in the Kathmandu district. In parallel, eight in-depth interviews were conducted with an interview guideline to collect the participants' perceptions and experiences of disability. Both quantitative and qualitative findings were integrated into the results. We found that about 77% of the participants with a physical disability had experienced depressive symptoms. Unemployment status (adjusted odds ratio (AOR) 2.7, 95% confidence interval (CI) 1.0–7.3) and comorbidity (AOR 2.5, 95% CI 1.0–6.0) had a statistically significant association with depressive symptoms. The majority of people with physical disabilities had negative experiences with societal prejudice and coping with their limitations. They were depressed as well as angry over having to stop their careers, education, and possibilities. Nevertheless, they were significantly happier and less sad than in their earlier years of life because of the possibilities, family environment, improved means of subsistence, therapeutic facilities, and supportive atmosphere at disability care homes. The policymakers should focus on preventing comorbidity and providing technical skills to persons with physical disabilities to improve their employment status and promote a healthy lifestyle.

**Funding:** The authors received no specific funding for this work.

**Competing interests:** The authors have declared that no competing interests exist.

## Introduction

The global burden of mental illness is projected to rise above $6 trillion by the year 2030 [1]. Depression is one of the most common mental disorders, affecting 300 million people worldwide and leading to suicide [2] and more than 0.8 million people die from suicide every year, and 75% of these occur in low- and middle-income countries (LMICs) [3]. Depression has become a significant public health problem with a constant increase in its prevalence. It is associated with different life experiences such as abuse, family and societal attitude, stress related to poverty, environmental barriers, lack of access to appropriate health care, etc., which are more common among persons with disabilities [4]. Depression contributes to the global burden of disease, predominantly becoming the single most important cause of years lost due to disability in middle and high-income countries. It is the third cause of disability worldwide, accounting for 4.3% of total disability-adjusted life years [5].

Persons with a physical disability are at least three times more likely to experience depressive symptoms as compared to the general population [1, 3]. Depression in persons with physical disabilities is found to be higher than in the non-disabled population [4]. Furthermore, physical disability intersects with gender [4] and ethnicity [6]. Physical disability has been defined by the World Health Organization (WHO) as "an existing difficulty in performing one or more activities which, in accordance with a person's age, sex and normative social role, are generally accepted as essential components of daily living, such as self-care, social relations, and economic activity" [7].

In Nepal, around 33% of people are suffering from mental health problems. Among these, over 90% of the population with mental health problems have no access to treatment [8]. Community-based epidemiological studies conducted in Nepal show that the prevalence of depression ranges from 28% to 41% [9]. In 2011, 1.94% of the total national population were persons with disabilities; among them, 36.3% had physical disabilities [10].

Despite its burden, mental health itself is a neglected and under-studied area of public health, particularly in LMICs [11]. It is even more neglected among persons with disabilities. In the Nepalese context, limited studies have been conducted to assess depression among the population with physical disabilities. Thus, we aimed to find out the prevalence of depressive symptoms and associated factors among persons with physical disabilities living in disability care homes in the Kathmandu district, Nepal.

## Methods

### Study design

We used a concurrent triangulation mixed methods study design where both quantitative and qualitative data were collected simultaneously and independently. This is a concurrent design where the quantitative method was dominant and the qualitative method was used for the triangulation of quantitative results in this study. The study aimed to determine the status of depressive symptoms and explore its associated factors among persons with physical disabilities living in care homes of Kathmandu District. The study was carried out from July to December 2018.

### Study population and sample size

At first, the lists of care homes for persons with disabilities were collected from the National Federation of Disabled-Nepal and then these care homes were approached for the data collection.

All care homes in the Kathmandu district provided a list of all individuals with physical disabilities. All persons with physical disabilities from disability care homes were included in the

sampling frame. Each person with a physical disability who was selected for sampling was taken as a study unit. The inclusion criteria of participants were limited to a single type of disability to avoid the complexities of multiple disabilities.

The required sample size was 162. Based on the formula, n = $z^2$pq/$e^2$, where $z$ is the standard normal variate (1.96) at 95% confidence interval, $p$ is the prevalence and 0.5 was used for the unknown prevalence [12], $q$ is 1—prevalence (i.e. 0.5), and $e$ is the margin of error (0.05), the sample size was 384. Further, using finite population correction methods, i.e., n = n/ (1+n/ N), where N = 256 (Total persons with physical disabilities in all the disability care homes of Kathmandu district were 256), we got 154 as a sample size for the study. Moreover, a 5% non-response rate was added to get the final sample size of 162.

The samples were selected by using a systematic random sampling technique. The list of all persons with physical disabilities was obtained from all the care homes of Kathmandu district and arranged alphabetically by care home names followed by the names of individuals. The first participant was selected by the lottery method. Then, each study participant was chosen using systematic random sampling from the arranged list where the sampling interval (K) was two, as N was 256 and n was 162.

A total of 162 participants were interviewed for the quantitative study. Eight persons with physical disabilities from among the same sample frame were selected purposively for the qualitative study, where in-depth interviews (IDIs) were conducted. Eight persons with disabilities were selected, considering that Manns et al. (2019) conducted the IDIs with eight respondents from non-profit organizations, a similar setting to our study [13].

## Quantitative data collection

A face-to-face interview was done with each participant to collect quantitative data from the representative sample. The validated tool of the Beck's Depression Inventory (BDI) scale was used in the study [14, 15]. The BDI is a 21-item, self-report rating inventory that measures characteristic attitudes and symptoms of depression [16] and has already been validated in the Nepali language [17] (S1 Text).

## Qualitative data collection

For the qualitative method, the principal investigator himself conducted in-depth interviews who had previous academic and professional experiences in conducting qualitative research and also had experience working for the rights of persons with disabilities. Similarly, IDIs were conducted with participants using the interview guideline. A face-to-face interview allowed the researcher to observe any non-verbal communication and also allowed both the interviewer and participant to seek any clarification necessary. Non-verbal communication was noted down during the time of the interview. Interviews were audio-recorded with permission from the participant to ascertain an accurate account of the interview, which can be replayed for analytic purposes and anonymity was assured during the course of the recording. We carried out the interviews on 15 days duration, i.e.,15th to 30th of September 2018.

## Data management and analysis

**Quantitative data.** The questionnaire was checked at the time of data collection for completeness of data. Data entry was done using Epi-Data version 3.1 and analysis was done in SPSS v 21 (IBM Corp., 2012. Armonk, NY). Quantitative data were expressed in the form of frequency and percentage. Individual's BDI score was categorized: 0–9 minimal depression, 10–16 mild depression, 17–29 moderate depression, and 30–63 severe depression [18]. Using logistic regression, we performed bivariate and multivariable analysis to observe the

association of independent variables, including the individual's socio-demographic characteristics, health behaviors, educational and employment status, with depressive symptoms. Independent variables, those found statistically significant (p value< 0.05) in bivariate analysis and important variables from the literature review, were further utilized in multivariable analysis. Multi-collinearity of variables was checked using the variance inflation factor (VIF). The highest VIF was 1.54, which suggested there was no multi-collinearity problem [19]. Statistical testing was done on a 95% significance level, where a p-value of less than 0.05 was considered to be significant.

**Qualitative data.** All the recordings of IDI were transcribed-translated, and verbatim were generated and compiled for coding. We used the thematic analysis method, as this method could be used in all types of qualitative studies in health science and patterns in the data that relate to participants' lived experiences, opinions and viewpoints, as well as behavior and practices, can be found by it [20]. We used Braun and Clarke's [21] six steps for doing thematic analysis, which involved "familiarizing with the data, creating initial codes, looking for themes among codes, reviewing themes, defining and labeling themes, and producing the final report". As part of the thematic analysis, final transcripts were coded by PK in tandem with another investigator (RB) with 75% inter-coder agreement in order to find similar patterns in transcripts. Afterwards, a consensus was built to finalize the codes through mutual understanding of coders and peer validation with a third researcher (BS). Each interview was summarized after coding with identified themes and categories of information. To generate meaning and further develop the themes, we looked over the textual data again. After analyzing quantitative and qualitative data, both results were combined and presented using convergent, divergent and expansive findings as per the concurrent triangulation design principles [22].

## Ethical consideration

Ethical approval for undertaking the study was obtained from the Institutional Review Committee (IRC) of the Institute of Medicine, Tribhuvan University (Ref: 178-075/76). The purpose of the study was clearly explained by the researcher to the administration of disability care homes and approval was taken before conducting the study. Considering the participants as persons with physical disabilities, written and verbal consent was taken from the participants before conducting the interview, and the interviewer ensured the right to terminate the interview at any time if s/he felt uncomfortable. The given consents were recorded in the questionnaire and then moved forward by the researcher. Also, verbal consent was recorded while recording the interview for the qualitative study which was approved by IRC. The participants, who were found to be severely depressed, were counseled by the interviewer and further informed to the administration of the disability care homes and referred to a nearby hospital.

## Results

### Quantitative findings

**Distribution of participants by their socio-demographic and health behavioral characteristics.** The participants' age ranged from 18 to 87 years, with a mean age of 47.5 ± 19.4 years. The socio-demographic characteristics of the study population are summarized in Table 1. Thirty-eight percent of the participants' families were dependent on agriculture, while 5.3% of the participants' families were involved in the service sector (monthly salary). At the same time, other families (38.8%) were fully dependent on the allowance provided by the government. More than 50% of the respondents were literate; among them, 13% had an informal education, and 2% had a bachelor's or above level of schooling. Among the participants, 48.8%

**Table 1. Distribution of participants by their socio-demographic and health behavioral characteristics.**

| Characteristics | Number | % |
|---|---|---|
| **Employment** | | |
| Self-employed | 42 | 25.9 |
| Unemployed | 120 | 74.1 |
| **Educational status** | | |
| Illiterate | 78 | 48.1 |
| Literate | 84 | 51.9 |
| **Gender** | | |
| Male | 90 | 55.6 |
| Female | 72 | 44.4 |
| **Marital status** | | |
| Married | 84 | 51.9 |
| Unmarried | 78 | 48.14 |
| **Family Type** | | |
| Nuclear | 122 | 75.0 |
| Joint | 26 | 16.0 |
| Single | 14 | 8.6 |
| **Family support** | | |
| Yes | 99 | 66 |
| No | 51 | 34 |
| **Comorbidity** | | |
| Yes | 79 | 48.8 |
| No | 83 | 51.2 |
| **Smoking** | | |
| Yes | 27 | 16.7 |
| No | 135 | 83.3 |
| **Alcohol drinking** | | |
| Yes | 26 | 16 |
| No | 136 | 84 |

had comorbidities. Major comorbidities were Gastritis, Hypertension, Arthritis and Asthma, among many others. Around 16.7% of the respondents were found to smoke cigarettes. Similarly, 16% of them were drinking alcohol.

**Distribution of participants by their nature of disability.** The nature of disabilities, use of assistive devices and the infrastructure are summarized in Table 2. Among the causes, 93.2% of participants became physically disabled after birth, either from accidents or from leprosy, while 6.8% had physical disabilities by birth.

Almost half (50.6%) of the participants used assistive devices such as wheelchairs, crutches, and prosthetic legs. The surrounding environment and the infrastructure of the disability care home were satisfactory to 89.5% of the participants.

Table 2 also shows the description of the participants based on the type of physical disability. We found most of the respondents (54.3%) had been affected by leprosy. Similarly, 26.5% of the respondents had a disability due to joint and spinal cord problems, while 4.3% had other types of physical disability, including stunting, muscular dystrophy and rickets.

**Distribution of participants by physical disability and depression-related characteristics.** Table 3 shows that the majority (77.4%) of the participants in the study had depressive symptoms. At the same time, 22.6% of them were found to have minimal disturbance or normal level of depression.

**Table 2. Distribution of participants by their nature of disability and types of disability.**

| Nature of disability | Number | % |
|---|---|---|
| By birth | 11 | 6.8 |
| After birth | 151 | 93.2 |
| **Use of assistive devices** | | |
| Use | 82 | 50.6 |
| Do not use | 80 | 49.4 |
| **Accessible infrastructure** | | |
| Satisfactory | 145 | 89.5 |
| Not satisfactory | 17 | 10.5 |
| **Type of disability** | **Number** | **%** |
| Leprosy-related impairment | 88 | 54.3 |
| Joints and spinal cord problem | 43 | 26.5 |
| Absence of body parts | 14 | 8.7 |
| Clubfeet | 7 | 4.3 |
| Polio | 3 | 1.9 |
| Others | 7 | 4.3 |

## Association of covariates with depressive symptoms

Table 4 describes the association of covariates with depressive symptoms. From the bivariate analysis (Crude Odds Ratio), the variables such as sex of participants, marital status, education status, alcohol drinking, employment status and comorbidity had a statistically significant association with depressive symptoms. After adjusting the variables, participants' employment status and comorbidity were found to be associated with depressive symptoms. The depressive symptom was more likely among participants who were unemployed [(AOR 2.7, 95% CI 1.0–7.3)]. Participants who had co- morbidities were more likely to have depressive symptoms [(AOR 2.5, 95% CI 1.0–6.0)].

## Qualitative findings

The age of interviewees ranged from 21 to 85 years. Interviewees' characteristics were illustrated in S1 Table. The findings from the interviewees were categorized into five major themes:

## Theme-1: Inaccessible geographic and social locations stimulate home stranding

Most of the interviewees said that they had their own home and family. They also wanted to live in their home along with their family members. However, they decided to live in a disability care home for various reasons. The most common reasons were mobility difficulties,

**Table 3. Distribution of participants by depression-related characteristics.**

| Level of depression | Number | Percentage (CI) |
|---|---|---|
| Normal or Minimal | 37 | 22.6 (16.2–29) |
| Mild Depressive symptoms | 67 | 41.4 (35.2–48.8) |
| Moderate depressive symptoms | 55 | 34.0 (26.5–39.5) |
| Severe depressive symptoms | 3 | 2.0 (0.0–4.3) |

CI = Confidence Interval (95%)

**Table 4. Association of covariates with depressive symptoms.**

| Variables | Unadjusted (COR) | 95% CI | p-value | Adjusted (AOR) | 95% CI | p-value |
|---|---|---|---|---|---|---|
| **Sex** | | | | | | |
| Female | Ref | | | Ref | | |
| Male | 2.2 | 1.0–4.9 | 0.043 | 0.4 | 0.1–1.0.4 | 0.075 |
| **Marital status** | | | | | | |
| Married | Ref | | | Ref | | |
| Unmarried | 2.5 | 1.2–5.4 | 0.013 | 2.4 | 0.9–6.4 | 0.06 |
| **Educational status** | | | | | | |
| Literate | Ref | | | Ref | | |
| Illiterate | 2.7 | 1.2–6.0 | 0.012 | 1.1 | 0.4–3.2 | 0.796 |
| **Employment** | | | | | | |
| Self employed | Ref | | | Ref | | |
| Unemployed | 3.4 | 1.5–7.4 | 0.002 | 2.7 | 1.0–7.3 | 0.047* |
| **Co morbidity** | | | | | | |
| Yes | 2.5 | 1.1–5.6 | 0.019 | 2.5 | 1.0–6.0 | 0.038* |
| No | Ref | | | Ref | | |
| **Alcohol drinking** | | | | | | |
| Yes | 2.5 | 1.0–6.1 | 0.043 | 0.8 | 0.2–2.5 | 0.726 |
| No | Ref | | | Ref | | |
| **Nature of disability** | | | | | | |
| By birth | Ref | | | Ref | | |
| After birth | 1.2 | 0.3–5.1 | 0.717 | 0.9 | 0.19–4.3 | 0.91 |
| **Use of assistive devices** | | | | | | |
| Use | 1.8 | 0.8–3.8 | 0.112 | 1.2 | 0.4–3.2 | 0.64 |
| Do not use | Ref | | | Ref | | |
| **Accessible infrastructure** | | | | | | |
| Satisfactory | 2.3 | 0.5–10.9 | 0.263 | 0.3 | 0.6–2.0 | 0.24 |
| Not satisfactory | Ref | | | Ref | | |
| **Smoking** | | | | | | |
| Yes | 1.5 | 0.6–3.8 | 0.360 | 0.6 | 0.2–2.0 | 0.48 |
| No | Ref | | | Ref | | |
| **Family support** | | | | | | |
| Yes | Ref | | | Ref | | |
| No | 0.6 | 0.2–1.4 | 0.304 | 0.7 | 0.2–1.9 | 0.52 |

AOR = Adjusted Odds Ratio, COR = Crude Odds Ratio, CI = Confidence Interval

* = Statistically significant at 95% CI

inaccessibility, unavailability of essential health services, interruption in continuing education and work, discrimination in society, and feeling a burden on family members.

**Sub-theme 1: Mobility difficulties.** Some of the interviewees who were not able to walk on themselves or needed to use a wheelchair mentioned those mobility difficulties due to the lack of wheelchair-friendly infrastructure around their homes and thus were obliged to leave home. Family members were required for their continuous care and support, making them feel burdened on the family.

*". . .Geographical structure of our village is not appropriate to use a wheelchair, there is no comfort toilet. . ."*- 35 years old, male, spinal-cord injury (P1)

**Sub-theme 2: Social stigma.** Some of the participants who were affected with leprosy; expressed that they had faced social discrimination leading to a feeling of humiliation and harassment—prevailing discrimination from society deprived them of various opportunities like education, work, social interaction and social support.

*"...In my community, people change their way if I [affected with leprosy] walk on the same road. I had to be discriminated against by my own society. So, I feel it is very difficult to return home now."* -63 years old, male, leprosy affected (P3)

People with physical disabilities received hate and discrimination not only from society, but they were challenged from their families as well. One of the participants said, *"...I am living here (care home), because my father used to scold me. He deprived me of education, sports and life skill opportunities against me..."* -21 years old, male, leprosy affected (P4)

**Sub-theme 3: Inaccessibility to health services.** Those having a disability due to spinal injuries needed frequent health services and support to maintain personal hygiene and other daily health care requirements. For those reasons, these interviewees preferred to stay in the disability care home.

*"...Many of us need to use a catheter, and we don't have those services in village..."* -35 years, male, spinal-cord injury (P1)

The behaviors and responses from society and relatives made persons with physical disabilities feel humiliated. All of these factors were associated with them leaving their homes and staying in a disability care home where they were able to find opportunities and friends with similar conditions to share their feelings and time with.

## Theme-2: Persons with disabilities are in jeopardy because of impoverished families

It was reported that the interviewees who had disabilities after accidents or injuries received financial support from their families initially, but the support was minimal and insufficient. Some participants' families were unable to provide financial support because of their lower economic status.

*"...My family is also from a low socioeconomic status. It is difficult for them to survive. So, how can they help...?"* -63 years old, male, leprosy affected (P3)

After living in disability care homes, most of the leprosy-affected participants had no support from their families. Moreover, persons with physical disabilities were concerned about their medical treatment if health issues arose, as their families were not in a position to support them with their out-of-pocket expenditures. Most of the persons with disabilities along with comorbidity experienced severe financial catastrophe in order to deal with their other medical complications in addition to the daily requirements.

*"...if I get sick, if I don't have money, my brothers are not going to help me any penny because they might think that I have nothing in my name and cannot get any benefit..."* -56 years, male, leprosy affected (P6)

Nevertheless, the provision of treatment and social security allowance from the government helped them live independently to some extent. They were able to get better food and shelter.

*". . .Doctor comes into the room and treats us, gives us care. The government gives us allowance by which we are living. . ."*-85 years, male, and leprosy affected (P8)

### Theme-3: Accepting and adopting bad and good environments of the care homes

Some of the participants who needed assistance during regular activities perceived that the response and behavior of their caretakers towards them were good.

*". . .If I need help or any assistance staff will help me. . ."*-25 years, female, spinal cord injury (P2)

Similarly, some of the participants reported adequate care and support from staff. Whereas few were reluctant to talk about the behavior of staff and caretakers. They deviated from this topic by saying,

*". . .We cannot say anything on this topic. . .We are living here somehow, it's good for us. . ."*-63 years, male, leprosy affected (P3)

**Sub-theme 1: Positive response.**   They felt the disability care home was comfortable because of the disability-friendly infrastructures where they can use a wheelchair everywhere (i.e., from the dining room, bedrooms, and washrooms).

*". . .Here, I can go everywhere in my wheelchair, from dining room to toilet and all. . ."*-25 years, female, spinal cord injury (P2)

All the participants expressed that the disability care home is like a home where they lived as a family. They responded that they could satisfy themselves by looking at each other because they were not the only ones and alone to have a disability.

*". . . It's good to stay here. All people affected with leprosy. . .stay here. . ."*- 71 years, male, leprosy affected (P5)

The participants said that various organizations also support their livelihood and they were able to receive vocational training, which helped them gain knowledge and skills and fruitfully spend their days.

*". . .After coming to this disability care home, I am taking training. I now feel that I can do something. I was a bit depressed. I came here and learned this job (making tools from bamboo). . ."*-35 years, male, spinal cord injury (P1)

**Sub-theme 2: Negative.**   Despite positive perceptions towards the environment of disability care homes, they experienced problems with the facilities and opportunities. The washrooms were inaccessible to the persons who were stunted and had prosthetic legs. They claimed that the restrooms were slippery and in unsanitary conditions.

"...*But toilets in this home is uncomfortable for me*..."- 34 years, female (P7)

"...*I feel good, but as I have a prosthetic leg, the toilet and bathroom is difficult for me*..."-21 *years old, male, leprosy affected (P4)*

There had been a few training sessions for persons with disabilities; however, some of the interviewees stated that the daily allowance provided in training was minimal.

"...*After coming in this care home, I am getting diverse training. I now feel that I can do something and I learn this job after coming here*..."- 35 years, male, spinal cord injury (P1)

"...*They [organization] gives us various training, we work the whole day, but we are not paid according to our work*..."- 34 years, female, stunted (P7)

A few participants were reluctant to answer about their perception of living in a disability care home. Participants expected more government-supported rehabilitation homes with beneficial vocational training.

**Sub-theme-3: Perceptions of using assistive devices.** Despite various benefits of using assistive devices, they had a more negative perception of the longevity and maintenance of the device. The maintenance cost of wheelchairs was prohibitive. Sometimes accessories for the wheelchair were challenging to find in the market, and they had to strive to get a new wheelchair.

"...*Umm, I use a wheelchair; it is difficult to use it. Sometimes it gets broken, its parts are difficult to find to mend it*..." -35 years, male, spinal cord injury (P1)

"...*We cannot afford to mend it every time, I save my pocket money to mend it if it's broken*..."-25 years, female, spinal cord injury (P2)

The wheelchairs they were using were donated by volunteer organizations or individuals. They also saved their limited pocket money for its maintenance. They were stressed, thinking the government has no policy provisions for distributing free assistive devices to people like them.

## Theme-4: Not being able to accomplish social tasks creates emotional distress

They seemed happy and usually accepted their disability after years of living along with other individuals with disabilities. But they were stressed and frustrated deep inside with the fact of having a disability. They felt humiliation, frustration, and stress when separated from their society and could not do routine tasks because of a disability. They shared that being physically disabled ruined their aims and deprived them of education and jobs. The participants were also concerned and worried about their children's future.

"...*After our death, my children won't be able to stay here. I am stressed because of my condition but more stressed because of the children's future*..." -56 years old, male, leprosy affected (P6)

Because of a disability, they had to live away from their families, and the dream of supporting their families was destroyed. The discrimination and ignorance from family and society made them sad. One of the participants planned to have a happy married life, which was unsuccessful because of being paralyzed.

*"...I am the elder son of my family, and my responsibility was to support the family, but I am unable now. In fact, I am seeking help from my family. I had dreams, but they were spoiled..." -35 years, male, spinal cord injury (P1)*

### Theme-5: Care homes are beneficial for the well-being of persons with physical disabilities

They found the disability care home to be comfortable and beneficial. The quality of life was increased after staying in the disability care home due to improved livelihood opportunities. All the people with physical disabilities were sharing one roof and had created and enjoyed their own wonderful world. The training provided by the care homes helped participants to be engaged the whole day fruitfully. Care homes provided organizational support and created a conducive environment for those who were literate where they were able to seek help from national and international supporting agencies to themselves and friends, regarding financial and other materialistic support by writing proposals.

*"...When I was living at my own home, I did nothing. I just stayed at home. Nobody cared about me. But here I am taking tailoring training. If I practice or learn it well, I can earn well in the market, which might help me economically..."-34 years old, female, stunted (P7)*

### Integrated findings

The results of quantitative and qualitative results were integrated into two domains, and they are presented below.

**Convergent findings.** The initial finding was that employment opportunities reduce depressive symptoms for persons with physical disabilities (Table 5). The quantitative results showed that unemployed persons with physical disabilities were more likely to have depressive

**Table 5. Integrated findings of employment status and comorbidity.**

| Variables /domains | Depression prevalence | | Qualitative result: IDI with diagnosed case of depression | | Triangulation of both significant results |
|---|---|---|---|---|---|
| | Yes % | No % | Experiences (problem faced) | Possible reasons | |
| **Unemployment** | 83.3 | 16.7 | *"...I applied for jobs that are also ruined now. Yes, I feel frustration sometimes. I think about the future and feel sad..."* (P1) *"Here, we can take training which is beneficial for us"* - (P7) *"...After I got training, I started earning some money for my living....I would have lived a discriminated life in my village. I have improved my quality of life after I came here"*- P(5) | Unemployed people are more vulnerable to economic problems, difficult life experiences and psychological problems as they face numerous barriers to employment. | 83.3% of the unemployed persons with disabilities had depressive symptoms. The odds of having depressive symptoms among unemployed participants were 2.7 times the odds of those compared to employed participants. The employment opportunity for persons with physical disabilities may reduce the depressive symptoms on them. |
| **Comorbidity** | 86.1 | 13.9 | *"...If I get sick, if I don't have money, my brothers are not going to help me any penny..."*- P6 *"I always regret why I was born... Huge financial problem for daily living...In addition, I got arthritis..."*- P8 | The qualitative findings suggested that persons with physical disabilities were worried about the treatment cost for their comorbidity, and pushed more toward the situation where they cannot work for their livings. Comorbidity is an additional burden to a person with a disability and may increase the severity of the condition, leading people into a depressive condition. | 86.1% of the participants with comorbidity had depressive symptoms. The odds of having depressive symptoms for persons with physical disabilities with comorbidity was 2.5 times the odds of those who do not have comorbidities. The occurrence of comorbidity among persons with physical disabilities may increase the probability of having depressive symptoms. |

symptoms than that of employed. Similarly, qualitative findings showed that persons with physical disabilities are eager to get jobs and become financially capable of running their life.

Another finding was that the presence of comorbidity among persons with physical disabilities might lead to a higher possibility of having depressive symptoms (Table 5). The quantitative finding suggested that persons with physical disabilities with comorbidity were more likely to get depressive symptoms compared to persons with physical disabilities without comorbidity. Moreover, findings of the qualitative study support that people with physical disabilities were feeling blue due to their other health conditions. They had to leave their homes and families to get regular medical services and they were worried that if they stayed back in villages, their health would have been worsened. In addition, being unable to accomplish social functions due to comorbidities caused them to have more distress.

**Divergent findings.**  Some of the qualitative findings opposed the findings from quantitative data analysis. The quantitative data analysis did not show an association for males and unmarried with depression; however, qualitative data showed that they were having distress. Nevertheless, the risk of depression was higher among unmarried [OR = 2.4 (95%CI = 0.9–6.4)] in quantitative analysis, although the risk was not statistically significant.

Likewise, qualitative findings showed that males had greater family responsibility and could not fulfil needs due to physical disability, which increases the discrimination and ignorance from family, by which they feel stress and humiliation.

Educational status was not found to be statistically significant in quantitative analysis, but those who were literate were able to seek help from relevant donors for financial support, capacity-building training and other necessary requirements for them and other members.

The quantitative analysis showed no association of depressive symptoms with family support. But qualitative study explores that many participants shared that low family support was the reason for the delay in treatment which leads to irreversible disability.

The use of assistive devices was not found to be statistically significant for depression in quantitative data analysis. But, the participants had a more negative perception on the longevity of the device. They often found it challenging to use a wheelchair because of the lack of wheelchair-friendly infrastructures. As some participants were economically unstable, they felt burdened to receive a wheelchair or other assistive devices without ensured maintenance from non-profit organizations.

In the qualitative study, most of the people with physical disabilities expressed that their overall quality of life has increased after residing in the care home. However, when probed if there were any challenges or negative aspects of living there, participants hesitated to talk about the institution where they were living.

## Discussion

In our study, more than two-thirds [i.e.,77.4% CI (69.1–84)] of the participants had depressive symptoms, which indicates physical disability can lead to depression [23]. This finding is consistent with the study conducted in Iran that showed 71% of the respondents had depression [24]. Similarly, a study conducted in Ethiopia showed the prevalence of depression among people with physical disabilities was 75.5% [25]. These findings are slightly higher than the studies conducted in Turkey (57.8%) [26] and Colombia (24%) [27]. These observed differences could be due to study design, such as sample size and measures of depression, as well as cultural differences.

### Convergent findings

In the bivariate analysis, gender, marital status, education, employment status and comorbidity condition of participants were associated with depressive symptoms. After adjusting all

possible covariates, employment status and comorbidity conditions were associated with depressive symptoms. Factors such as gender, employment and education are a sign of social status, which may have an impact on mental health [28]. This study illustrates that unemployed persons with a physical disability were likely to have a higher chance of getting depressive symptoms. Participants reported minimal financial support from their families. Unemployed people were more vulnerable to economic problems, difficult life experiences and psychological problems as they face numerous barriers to employment [29]. At the same time, employment of persons with disabilities builds the self-confidence and motivation required to foster higher challenges [30]. In our study, both quantitative and qualitative findings showed employment status of participants was significantly associated with depressive symptoms. Similar findings were reported in the study conducted in Mexico, the United Kingdom [31] and Korea [32], which reflects that unemployed participants are more likely to have depressive symptoms.

Our study evidenced that participants with additional health conditions were significantly associated with depressive symptoms. This might be because the financial burden faced by being disabled is exaggerated by out-of-pocket expenditure due to treatment of additional health conditions. This finding is supported by the study conducted in Korea, which shows a significant linear association between depression and comorbidity [33]. Our qualitative study showed that the potential dependence due to various comorbidity and the burden of its management that requires others' support and financial problems causes increased stress to persons with physical disabilities, further leading to depression. Comorbidity is an additional burden to a person with a disability and may increase the severity of the condition, leading people into a depressive condition. A study conducted among adult males with lower limb amputation also found an association between socioeconomic status, society's attitude towards disability and comorbidity with depression. Rather than physical disability, low financial status was the major reason for experiencing depressive symptoms [34].

## Divergent findings

In our study, the gender of participants was not associated with depressive symptoms in multivariate analysis. But the qualitative study showed that males have greater family responsibility and cannot fulfil needs due to physical disability, which increases the decimation and ignorance from family, by which they feel stress and humiliation. However, the result of a longitudinal study conducted in Canada showed both males and females with a disability had a higher risk of depression [7]. A study conducted in Jordan showed that unmarried adults with disabilities were more likely to experience depression. Nonetheless, our study showed no significant association between marital status and depressive symptoms among persons with physical disabilities after adjusting all the possible covariates. But unmarried persons with disabilities were lonelier as they were unable to share their feelings with anyone.

This study showed no association of depressive symptoms with family support. But qualitative study explores that many participants shared that low family support was the reason for the delay in treatment which leads to irreversible disability. In addition, a study conducted in India found that people with physical disabilities have higher levels of depressive symptoms than non-disabled people. At the same time, they feel much more frustrated and helpless and do not fit with ordinary people in society. The study also found a positive correlation between a low level of self-esteem and a high level of depression among persons with physical disabilities [35]. Also, this study found that having a physical disability contributed to experiencing depressive symptoms. Participants' experienced societal discrimination, mobility difficulties,

lack of family support, and being deprived of various opportunities after being affected by a disability had a cumulative effect on experiencing depressive symptoms.

A study conducted in the United States, using a large population-based sample, found the prevalence of current depressive syndrome (CDS) higher among adults with physical disabilities who used assistive technology (AT) than those who did not use any AT. Similarly, the prevalence of CDS among AT users was about three times higher than among AT non-users [36]. In contrast, the use of assistive devices was not found to be statistically significant in our study. The participants had a more negative perception of the longevity of the device. They often found it challenging to use a wheelchair because of the lack of wheelchair-friendly infrastructures. Though some of the organizations supported assistive devices, it was very difficult to manage the costs even for their maintenance. Thus, they were stressed that the government had not taken any steps to provide support on assistive devices.

Though the majority said that the environment was good in the institute, many of them were found to have depressive symptoms. This could be due to the Hawthorne effect, as they were residing there and were reluctant to say bad about the institute. A study conducted among adults with physical disabilities in China found that a physical disability hindered their basic activities of daily living, and were profoundly depressed, further resulting in more functional limitations [37].

### Expansive findings

In general, the factors like mobility restrictions, social stigma, and inaccessibility to health services displaced persons with physical disabilities from their homes and communities as it exaggerated their frustrations. In addition, from the qualitative study we found that the care homes provided the basic needs required in their daily life, which was not readily available at their home. Furthermore, care homes provided infrastructure, government and other non-governmental support at institutions, and training, among many others.

### Limitations of the study

Our study did not include persons with multiple disabilities to reduce the non-response rate because of the severity and lack of specialized human resources to interview those with deaf, deaf-blind, or other disabilities. In addition, the perception of caretakers and families was not included; therefore, their impression is unknown. As the qualitative analysis was done independently in our study, interviews with only eight persons with physical disabilities may not be sufficient to reach saturation. Also, this study is institution-based rather than community-based, which might have affected the depression status of persons with physical disabilities positively or negatively as they were living in an enabling environment but far away from their families.

### Conclusion

Our study revealed that more than three-quarters of the persons with physical disabilities had depressive symptoms, where employment and comorbidity status were strongly associated with depressive symptoms among them. The findings indicate the major factor behind the depressive symptoms could be comorbidity and financial dependence. Also, qualitative findings showed that males and persons with physical disabilities without family support were more likely to be stressed due to social roles. Based on the results, we would like to suggest authorities for the provision of occupational therapy and psychotherapy along with vocational and life skills training to improve their quality of life.

## Supporting information

**S1 Text. Questionnaire.**
(DOCX)

**S1 Data. Data.**
(XLSX)

**S1 Table. Participants details.**
(DOCX)

## Acknowledgments

Special thanks to our research participants, authorities who allowed us to conduct our study, and teachers who guided us. No author had any financial or personal conflict of interest among the authors supporting the research.

## Author Contributions

**Conceptualization:** Prabin Karki, Binjwala Shrestha.

**Data curation:** Prabin Karki, Nabin Adhikari.

**Formal analysis:** Prabin Karki, Prasant Vikram Shahi, Rabindra Bhandari, Nabin Adhikari.

**Methodology:** Prabin Karki, Prasant Vikram Shahi, Krishna Prasad Sapkota, Rabindra Bhandari, Nabin Adhikari, Binjwala Shrestha.

**Project administration:** Prabin Karki.

**Software:** Prabin Karki.

**Supervision:** Prasant Vikram Shahi, Krishna Prasad Sapkota, Rabindra Bhandari, Nabin Adhikari, Binjwala Shrestha.

**Validation:** Krishna Prasad Sapkota, Binjwala Shrestha.

**Writing – original draft:** Prabin Karki.

**Writing – review & editing:** Prabin Karki, Prasant Vikram Shahi, Krishna Prasad Sapkota, Rabindra Bhandari, Nabin Adhikari, Binjwala Shrestha.

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
