## [Decision Letter · Decision Letter 0]

9 Aug 2022

PGPH-D-22-01140

Depressive symptoms and associated factors among persons with physical disability of Kathmandu district, Nepal: A mixed method study

Dear Mr. Prabin,

Thank you for submitting your manuscript to PLOS Global Public Health. After careful consideration, I feel that it has merit but does not fully meet PLOS Global Public Health’s publication criteria as it currently stands. Therefore, I invite you to submit a revised version of the manuscript that addresses the points raised during the review process.

We look forward to receiving your revised manuscript.

Kind regards,

Rakesh Singh

Academic Editor

Journal Requirements:

1. In the ethics statement in the Methods, you have specified that verbal consent was obtained. Please provide additional details regarding how this consent was documented and witnessed, and state whether this was approved by the IRB.

2. Please update your online Competing Interests statement. If you have no competing interests to declare, please state: “The authors have declared that no competing interests exist.”

3. Please provide a complete Data Availability Statement in the submission form, ensuring you include all necessary access information or a reason for why you are unable to make your data freely accessible. If your research concerns only data provided within your submission, please write “All data are in the manuscript and/or supporting information files.” as your Data Availability Statement.

4. Please ensure that you refer to Table 2 in your text as, if accepted, production will need this reference to link the reader to the table.

5. We notice that your supplementary materials are included in the manuscript file. Please remove them and upload them with the file type 'Supporting Information'. Please ensure that each Supporting Information file has a legend listed in the manuscript after the references list.

Reviewers' comments:

Reviewer #1: The article is about a very important issue and is very interesting as well. However, it does lack clarity in some places. Hence, some modifications are needed to make the article better.

Suggestions:

There are grammatical errors in some places, please proofread the document.

Line 89-95:

How is this a “mixed explanatory concurrent study“?

The information in the methodology part is confusing. If it is explanatory, will the study be concurrent or sequential? Are you trying to triangulate findings or explain reasons from quantitative studies?

Line 111: How was Systematic Random Sampling used?

Line 112: Why were people more than sample size taken?

Line 151: Any reason for not taking written consent?

Line 175: Do you mean 0.5 instead of 10.5 %? Check for similar errors?

Table 3: Please write what the numbers in parentheses indicate.

Line 208: Why were more male participants chosen for qualitative study? Any specific reason for that?

Line 465: Why were people with multiple disability excluded? Don't you think this may affect the findings? Was this in the eligibility criteria?

References: Some references are not according to the journal guidelines. Please make changes accordingly.

Reviewer #2: Title: The study is done in the care homes so how can represent the persons with physical disability of Kathmandu district?

Line 31: This design is not correct. Explanatory design must be sequential and concurrent design must be used to triangulate the findings from qualitative and quantitative findings. Also describe which method was dominant.

Line 33: Does BDI give symptoms only or more? It is used in clinical practice as well so make sure you have used the right tool to measure the depressive symptoms. DASS is used more frequently to measure the symptoms and BDI is used for diagnosis.

Line 36: This is a classical concurrent mixed method. It is not a sequential design.

Line 91: Make sure that this is the right tool to investigate depressive symptoms or not as it is also widely used to diagnose depression.

Line 93: No. It was collected simultaneously as described above so this was not based on the sequential mixed method design. Change this accordingly

Line 97-101: Change the title as per this description, do not over generalize. Title should represent what was done and where it was done and to whom it was done.

Line 105: Write “minimum sample size”.

Line 106: Why was this proportion used? Provide justification.

Line 120: Change: … scale in Nepal language was …

Line 125: Describe clearly who did this and who conducted the interview. How much experience did the interviewer have for conducting this type of interview?

Line 131: Are you referring to the code book? If yes then modify accordingly. Coding is not in the qualitative data!

Line 132-133: Provide reference of both the software. If you have license of the SPSS software mention it too so that it becomes ethical to use it for this research.

Line 135-37: Provide reference here.

Line 137-40: How did you decide to take independent variables from bivariate analysis to multivariate analysis. Did you check for confounding using variance inflation factor? If yes then provide how it was used along with the cut-off used for this particular regression model.

Line 141: Did you use this for taking the variables from bivariate to multivariate model too? If yes, provide justifications.

Line 142-46: Which analysis framework was used. How did you ensure the reliability of the coding? Write about the researcher’s reflexivity here.

Line 168: Why only? Write Around here as you are not comparing.

Line 193: What are you presenting in the parentheses here in Table 3?

Line 204: Confounding effect among the factors eligible for the final model was not done so the results presented in Table 4 are not valid.

Line 207: Describe the analysis framework used e.g. conceptual content analysis or relational content analysis. If it was conceptual content analysis then describe which framework was used to analyze the data and arrive at the themes e.g. Braun and Clarke’s six steps for doing thematic analysis.

Line 212: Describe the process of open coding followed by axial coding and selective coding process used to reach this theme.

Line 371: Organize this as convergent finding, divergent finding and expansive finding.

Line 388: Present these results in terms of convergent findings, divergent finding and expansive findings. Read how to mix/triangulate concurrent mixed methods result and do the needful.

Line 390: Present these results in terms of convergent findings, divergent finding and expansive findings. Read how to mix/triangulate concurrent mixed methods result and do the needful.

Line 393: Re-write the discussion based on convergent, divergent and expansive findings that will be obtained after you re-analyze your data accordingly.

Line 465-467: You study is not community based; it is institutions based so highlight it too here.

Line 468: Re-write the conclusion based on convergent, divergent and expansive findings that will be obtained after you re-analyze your data accordingly.

---

## [Decision Letter · Decision Letter 1]

4 Nov 2022

PGPH-D-22-01140R1

Depressive symptoms and associated factors among persons with physical disabilities in disability care homes of Kathmandu district, Nepal: A mixed method stud

Dear Dr. Karki,

Thank you for submitting your revised manuscript to PLOS Global Public Health. After careful consideration, we feel that it has improved a lot and has merit but does not fully meet PLOS Global Public Health’s publication criteria as it currently stands. Therefore, we invite you to submit a revised version of the manuscript that addresses all the points raised during the review process, more specifically Reviewer 2. The manuscript still required work from the authors to make it technically sound. There is an issue with multicollinearity (confounding) test in the regression model and only 8 interviews were done for the qualitative part of the study. Also the authors have not presented the mixed methods finding as per the design and discussion and conclusion are also not done accordingly. 

We look forward to receiving your revised manuscript.

Kind regards,

Rakesh Singh

Academic Editor

Reviewers' comments:

Reviewer #1: The authors have addressed most of the previous suggestions efficiently. Hence, the quality of the manuscript has improved considerably. However, there are some minor changes that can make the manuscript even better.

Suggestions:

• There are still some grammatical and language errors. Please read the document thoroughly.

• Keywords: Should be in alphabetical order

• Table 3: Is this 95 % confidence interval?

• Table 4: Write the meaning of asterisk (*) below the table

Reviewer #2: Line 38/39 - This result must be verified after ruling out the multicollinearity among the independent variables.

Line 95 - use method instead of methods here as you have used only one type of qualitative method

Line 97 - remove the name of the tool used, not required here

Line 97 - replace investigate with proper action verbs for qualitative and qualitative methods

Line 98 - replace in with 'living in care homes of'

Line 103 - It is not clear why other districts are mentioned here, see comments in the manuscript and do accordingly

Line 109 - Sample size is not correct, need re-checking

Line 109 - Margin of error is represented by 'e' so use e instead of d in the formula

Line 110 - Reference or justification is required for p=0.5

Line 111 - Replace d=allowable error with e=margin of error

Line 111 - Value of N in the Finite population correction is missing, you must provide it with proper reference

Line 116 - Describe how the list was arranged (see comment in the manuscript for details)

Line 117 - Determine how the sampling interval was determined i.e. provide information on n and N used

Line 119 - Eight person is not enough to get saturation for the qualitative study as the qualitative analysis is done independently in the concurrent mixed methods design

Line 120 - Why "randomly"? Provide justification

Line 121 - So, in-depth interviews was the only qualitative method used in this study

Line 146 - Add standard deviation after mean

Line 152 - Replace "uni-variate" with "bi-variate"

Line 153 - VIF values is not correct for the logistic regression model

Line 156-57 - The p-value for bi-variate model is different than the multivariate model so change accordingly (see the manuscript for the detail comment)

Line 166-67 - Report interceder reliability index here

Line 168 - Describe the code learning process, how many were retained and the final interceder agreement

Line 172 - No! As per your design, you must present your findings in terms of convergent, divergent and expansive findings

Line 241 - No! You can't use the final model before assessing the multicollinearity so do it

Line 247 - Results are biased as multicollinearity was not assessed before using the final model

Line 420 - No! You must present your results in three sections here: convergent findings, divergent findings and expansive findings

Line 438 - You must compare and contrast the findings from convergent, divergent and expansive findings here

Line 533 - You must conclude based on the findings from convergent, divergent and expansive findings.

---

## [Decision Letter · Decision Letter 2]

16 Dec 2022

Depressive symptoms and associated factors among persons with physical disabilities in disability care homes of Kathmandu district, Nepal: A mixed method study

PGPH-D-22-01140R2

Dear Mr Karki,

I am pleased to inform you that your manuscript 'Depressive symptoms and associated factors among persons with physical disabilities in disability care homes of Kathmandu district, Nepal: A mixed method study' has been provisionally accepted for publication in PLOS Global Public Health.

Best regards,

Rakesh Singh

Academic Editor
